# The Economic Burden of Chromosome Translocations and the Benefits of Enhanced Screening for Cattle Breeding

**DOI:** 10.3390/ani12151982

**Published:** 2022-08-05

**Authors:** Nicole M. Lewis, Carla Canedo-Ribeiro, Claudia C. Rathje, Rebecca L. Jennings, Maxim Danihel, Lisa M. Bosman, Giuseppe Silvestri, Darren K. Griffin

**Affiliations:** School of Biosciences, University of Kent, Canterbury CT2 7NH, UK

**Keywords:** reciprocal translocation, Robertsonian translocation, 1:29, chromosome, cytogenetic screening, FISH, fertility, AI, financial model

## Abstract

**Simple Summary:**

The global cattle industry, dairy and beef, provides 81% of the world’s milk and 22% of its meat requirements. Milk production has more than doubled in the last 50 years, and there is increased focus on increasing production by more efficient, sustainable means. A key to this success is the bull, which, if sub-fertile, can have a severe impact on costs and the environment. The leading cause of bull infertility is so called “RTs”, where large parts of the genome are translocated to other parts where they would not normally reside. The purpose of this study was to provide a framework for calculating the financial impact that an RT can have, and therefore the benefits of a screening programme to not using such bulls for breeding. We recently developed a novel method of RT screening, which we think detects at least three times as many compared to the traditional method. We calculated that the economic benefit of proactively screening (and therefore not using) a bull with an RT could be GBP 7.2 million pounds (nearly USD 9 million) per bull over six years. Our expanding knowledge of the incidence of genetic abnormalities and their associated costs to production support the decision of the cattle industry to use screening approaches to guard against the use of bulls with RTs.

**Abstract:**

The cattle breeding industry, through both of its derivatives (dairy and beef), provides 81% of milk and 22% of meat required globally. If a breeding bull is sub-fertile, this impacts herd conception and birth rates, and it is generally accepted that having a proactive genetic screening programme can prevent further losses. Chromosome translocations are the leading genetic cause of infertility in livestock and, in cattle, this extends beyond the classical 1:29 to other Robertsonian translocations (RobTs) and to reciprocal translocations (RECTs). The incidence of both (collectively termed RTs) varies between breeds and herds; however, we estimate that RECTs are, most likely, at least twice as common as RobTs. The purpose of this study was to develop an industry economic model to estimate the financial impact of an RT event at the herd level. If we assume a conservative incidence rate of 0.4% for Rob1:29 with each one impacting the conception rate by 5%, we calculate that actively screening for and removing a Rob1:29 bull could benefit an impacted herd by GBP 2.3 million (approx. USD 2.8 million) over six years. A recently updated screening protocol developed in our lab for all RTs, however (with a projected combined incidence of 1.2%, impacting conception rates by 10%), could benefit an impacted herd by GBP 7.2 million (nearly USD 9 million) for each RT found. For an industry worth USD 827.4 billion (dairy) and USD 467.7 billion (beef), expanding knowledge on incidence and further dissection of the potential costs (financial and environmental) from RTs is essential to prevent further losses.

## 1. Introduction

The global cattle industry, through both of its derivatives (dairy and beef), provides 81% of the world’s milk and 22% of the world’s meat needs [1]. Cow milk production has more than doubled in the past 50 years, now at more than 800 million tonnes per year, and is predicted to increase with the growing population [1]. Across commercial livestock breeding programs, breeders in both the dairy and beef industry focus their objectives on increasing output and efficiency, which, in addition to maximising profits, can also reduce waste and environmental impact per animal produced [2]. Optimal reproductive strategies are essential to this process and, as such, sub-fertile animals can have a significant negative impact on productivity. This is especially true of males where the use of artificial insemination (AI) for reproduction in cattle has become widely adopted, particularly in the dairy segments [3]. Because bulls can produce large quantities of semen (around 800–1200 doses per week), the industry relies on a small pool of high value animals that have been selected for beneficial genetic traits, but not necessarily for fertility [3,4]. This is becoming increasingly relevant with regard to not only dairy production, as the combination of AI with sexed semen is becoming more widely used, but also for beef, where the benefits of using AI over natural service are being realised [5]. On average, a breeding bull can inseminate from around 130,000 to 200,000 cows in its lifetime and, in extreme cases, this can rise as high as 400,000 [6].

If a bull is sub-fertile, this can impact the conception and birth rates; it can also pass on any genetic abnormalities to his offspring, affecting their fertility and thereby perpetuating further opportunities for cost and economic losses for the producer. It is generally accepted that having a proactive screening programme for genetically determined fertility issues in bulls can help prevent further losses. Nonetheless, the extent to which genetic abnormalities causing infertility are a financial burden to the modern cattle breeding industry is hitherto not calculated. Consequently, the likely benefits of a screening programme have not been fully appreciated, nor has the additional benefit of a system that identifies additional abnormalities not detected by older methods.

Balanced chromosomal rearrangements are the leading genetic cause of fertility issues in livestock, recently reassessed in the pig breeding industry as being present in around 1% of breeding boars when enhanced screening approaches are used [7,8]. We recently calculated the financial impact on the industry of a translocation-carrying boar as between GBP 69,802 and GBP 51,215,378 per animal, depending on its position in the breeding pyramid [8]. In pigs, most chromosomes in the karyotype are found translocated; however, among bulls, the distribution is non-random. The karyotype of domestic cattle (both *Bos indicus and Bos taurus*) comprises 29 pairs of acrocentric autosomes (the submetacentric X and the tiny Y completing the set), making Robertsonian translocations (centric fusions) very likely [9]. Indeed, arguably the best known non-human chromosome abnormality in all of the scientific literature is a Robertsonian translocation (RobT) of the largest and smallest cattle autosomes (1:29). This specific abnormality has been observed in over 50 breeds around the world, with reported frequencies varying from up to 60% to as low as 0.4% in Holsteins [10]. As with any balanced translocation, the phenotypic effects are rarely seen in the animal itself, but manifested in its subsequent reproductive performance, largely because of perturbations of the normal meiotic process, leading to a reduction in the number of gametes produced, and chromosomal unbalanced products that can lead to implantation failure and/or pregnancy loss [11].

In addition to the 1:29 RobT, over 44 other similar errors (also RobTs) have been observed and described in cattle, with equivalent negative effects on fertility [9]. These additional RobTs have a lower reported incidence rate: in general, around 6.5% in Holstein cattle; however, this number varies with other breeds [10]. It is possible, however, given the difficulty in identifying individual chromosomes, that some RobTs are identified as 1:29, when in fact they involve other chromosomes. RobTs, in general terms, are nonetheless relatively easy to detect with standard Giemsa staining (with or without banded karyotyping), as the two fused chromosomes will appear as one larger, (sub)metacentric chromosome [9]. Reciprocal translocations (RECTs) are, however, far more difficult to identify as not only are all the autosomes acrocentric but they are all of similar size. As such, while, with relatively sophisticated extended chromosome preparation and G-banding techniques, RECTs can be identified, it is estimated that only 16% are spotted, leaving 84% undiagnosed [11]. In point of fact, most labs only stain chromosomes homogeneously (i.e., without banding) in order to spot 1:29 and other RobTs, and do not attempt to identify RECTs.

Finding a precise figure as to the incidence of RECTs in cattle is not an easy task, mostly because most labs do not look for them. Moreover, finding a relative incidence compared to RobTs is complicated by the fact that the incidence of RobTs is so variable, in turn depending on the extent to which screening programmes have near eliminated them. De Lorenzi et al. [12] suggested that the rate of RECTs is around four–five times the rate of non-1:29 RobTs; however, if we accept that at best, only 16% were identified [11], it seems clear that RECTs are an under-discovered genetic hazard, at least as significant as the 1:29 RobT, and possibly much more so. One of the purposes of this study was to estimate the relative rates of RECTs vs. RobTs.

Cattle breeding companies are often required to demonstrate that their breeding bulls are free of the 1:29 RobT [13]. The rationale for this is clear in that the risks of letting an undiagnosed translocation into the breeding herd would have significant costs in terms of finance, loss of reputation and the environment. Pregnancy rates per insemination in UK herds are in the region of 50–60% and it takes at least 55 days to establish whether a cow is pregnant [1], which leads to a loss of productive days lactating if she is not. The effect of RobT 1:29 are therefore profound. By extension, therefore, the same would apply to other RobTs and RECTs. Tests to detect RobT 1:29 are often (by PCR or FISH) targeted at this translocation alone, rather than using standard chromosome preparations [13], which would at least detect other RobTs. Thus, if employing standard chromosome preparation techniques, a reasonably competent laboratory would detect a RobT that involved chromosomes other than 1 and 29; however, it would be unlikely to detect an RECT at all.

In recent years, we developed a screening protocol that can readily detect both RobTs and RECTs (collectively henceforth referred to as “RTs”) for all cattle chromosomes [3]. The method utilises fluorescence in situ hybridisation (FISH), sub-telomeric probes and a two-slide multi-hybridization device, with probes for distal (furthest from centromere) and proximal (nearest) regions, one per chromosome, per square, highlighting the chromosome of interest. RECTs are detected by probes appearing on different chromosomes and RobT fusions are also easily spotted [3]. This, in turn, is based on our current pig-based screening approach [7] which has, to date, been used to screen over 2500 boars in our laboratory. While we have calculated the potential financial impact of a pig chromosome translocation entering the breeding herd [8], we have yet to do the same for the (dairy) cattle breeding industry. The purpose of this study is to achieve this. Moreover, we provide an update of our efforts with the cattle device, improving on the prototype previously reported in [3] and describing new, hitherto unreported chromosome abnormalities.

## 2. Materials and Methods

### 2.1. Cytogenetic Preparation

All preparations used in this study were from heparinised whole blood samples collected as part of standard procedures used for commercial evaluation by in house trained veterinarians. As all bloods were collected for routine testing, no further ethical approval or oversight was deemed necessary. All bulls were between two and six years old and consisted of Holstein, Jersey Belgian White and British White Breeds.

Cultured for 72 h using PB MAX Karyotyping medium (Gibco, Thermo Fisher Scientific, Waltham, MA, USA) at 37 °C, 5% CO_2_. Cell division was arrested by adding 10.0 g/mL colcemid (Gibco, Thermo Fisher Scientific, Waltham, MA, USA) for 35 min before hypotonic treatment with 0.075 M KCl and fixation to glass slides using 3:1 methanol:acetic acid.

### 2.2. Fluorescent In situ Hybridization

The cattle chromosome screening device was based on our previous work [3] with modifications. Fluorescent-labelled probes (fluorescein labelled for proximal (near to centromere), Texas Red for distally located probes) at a concentration of 10 ng/µL were dried on a proprietary Chromoprobe Multiprobe System device (Cytocell Ltd., Cambridge, UK); 24 squares for chromosomes 1–24, and 8 squares for chromosomes 25–29 + X. Fixed metaphase preparations were placed on each square of the glass slide, which was dried, then washed in 2× sodium saline citrate (2 × SSC) and dehydrated using an ethanol series. Hybridisation buffer (Cytocell Hyb I, Cambridge, UK) rehydrated the dried probes on the counterpart device, with slide and counterpart sandwiched together, warmed on a 37 °C hotplate (10 min) denatured at 75 °C (5 min) then hybridised overnight at 37 °C. Post hybridisation washes were 2 min in 0.4 × SSC (72 °C) 30 s in 2 × SSC/0.05% Tween 20 at room temperature. Chromosomes were counterstained using DAPI (blue fluorochrome) in VECTASHIELD anti-fade media and viewed using an Olympus BX61 epifluorescence microscope (Olympus, Tokyo, Japan) and SmartCapture (Digital Scientific UK, Cambridge, UK) system. Modifications from [3] included improved probe selection and the use of proprietary labelling techniques (Cytocell, details not disclosed). For each square of the Chromoprobe Multiprobe System, 3–5 metaphases were analysed.

### 2.3. Calculations of the Benefits of Chromosome Screening

To then understand the nature of the cattle breeding industry and, in particular, how an improved chromosomal screening approach might be a substantial improvement on Giemsa chromosome staining, an approach mirroring that of our previous study [8] was used. Specifically, we examined the UK industry, focusing on the effects relating to genetics companies, bull stud businesses, pure dairy farms, dairy and beef combined, as well as pure beef businesses. Sources for raw figures included *Farmers Weekly*, published journals, government websites such as AHDB and the FAO (e.g., [5,6]), and individual interviews with key players in the industry. Genetics companies for the dairy and beef sector typically have genetic breeding programs for bulls, where they are selected based on breed characteristics and individual desired traits. Examples of these traits include milk production traits, fat index, reproduction, and longevity [14]. Studs not only breed and maintain male animals, but also collect and sell the semen (which may or may not be sexed by flow cytometry). This is particularly relevant for the dairy industry as one male would typically be responsible for inseminating numerous female dairy cattle [1].

Because cattle are not litter-bearing animals, unlike our previous work in pigs [8], we focussed our model on the effects of a chromosome on the herd. Moreover, in the pig breeding industry, a relatively rigid pyramid structure exists, including dam line and nucleus great grandparents (GGP), multiplier grandparents (GP), and commercial parents (P) of the slaughter generation. By contrast, the cattle (both dairy and beef) are conceptually simpler. Genetics companies sell semen to producers; the average healthy bull, if collected twice per week, can create 800–1200 straws of semen per week (1600–62,400 straws per year [4]). It takes an average of 2.2 straws to inseminate a cow [4], which gives between 18,900 and 28,363 doses (or possible matings) per year, per bull. For this reason, an average of 23,000 matings per year was imputed into a financial model designed, ultimately, to calculate the benefits of chromosome screening.

In order to demonstrate such benefits, it is necessary to establish the potential economic hazards of a bull with an RT to the herd. It has been previously reported that a calving rate of 50–60% is reduced by 5–10% by each Rob1:29 translocation [3,8,13,15] and, thus, a conservative estimate of a reduction in calving rate from 55% to 50% was first used in our calculations. Through this model, we expressed results as the number of cows not impregnated and thus likely not lactating (see results and Table 2). As it takes approximately 55 days until another attempt at insemination can be made, the next outcome of our analysis was calculated as the number of days without milk that occurs normally, compared to the scenario when a bull with a translocation was used to inseminate the herd. This figure was then used as a basis to calculate the value of the milk lost, plus the value of the calf that could be sold and thereby the total opportunity loss of an RT per year to a herd (Table 2). The “cost per year” (i.e., the amount that the producer loses out on an opportunity and must pay for extra nutrition for the cow) was calculated by looking at how much the producer would make per day, based on the difference in value between an extra day at peak milking time verses an extra day at the end of the lactation. General housing costs were not taken into consideration because the cow would need to be housed regardless; however, the extra costs come from extra feed and nutrition to produce extra milk, or margin over purchased feed, which was assumed to be GBP 0.20 per litre of milk produced. The calculation was a complex one based on intensive interviews with leading players in the industry. In essence, however, starting with a calving index of 13 months, 4 months were subtracted from this to account for dry and non-peak times for milking, giving a value of 9 months. Next, looking at an estimate of the persistency of lactation, cows milked three times per day will, on average, lose 6% of their peak milk per month and (using a peak of 50 L and the 9-month value from above) gave a reduction in milk 54% or 27 L. The other opportunity cost is the value of the calf that could be sold. This was established by dividing the average value of the calf by the calving interval. To calculate the benefits of screening for RTs, the opportunity cost (per year from days without milk) was calculated at 55% (no RT present), 50% (a reduction in conception rate of 5% as a result of mating with a Rob1:29 bull, and also 45%—a reduction of 10% (rationale given in results section). The relative benefits of screening were then expressed as the difference between these opportunity costs. In order to establish the relative benefits of using our own screening approach (3 and reported herein), it was necessary to take into account the relative incidence of RECTs and other (non-1:29) RobTs, the greater impact if RECTs over RobTs, then, finally, the relative cost of our own test compared to standard chromosome analyses.

## 3. Results

### 3.1. Chromosome Screening

To date, using our device, we have screened 59 bulls of four breeds, identifying 7 unrelated (to best of knowledge) RobT (2 British White, 5 Holstein), 3 RECT (Holstein), 1 complex translocation, and 1 sex chromosome chimeric bull (both Holstein; figures also include those previously reported [3]). A full breakdown of these findings is presented in Table 1, while an example of a RECT discovered by FISH (and which would have likely escaped detection by conventional karyotyping) is presented in Figure 1.

### 3.2. Financial Effects of an RT on a Herd

As mentioned in the materials and methods, the normal herd conception rate in this model was considered to be 55% and we considered the impact of a herd inseminated by a bull with an RT to be the conservative estimate of a reduction of 5% (though the literature quotes 5–10%) to 50%. Using those rates, from 23,000 matings (see materials and methods), a bull without such an abnormality would yield 12,650, while a bull with an RT would yield only 11,500 successful births. This would mean that approximately 10,350 (without an RT) and 11,500 (with RT) cows would not be impregnated and, thus, likely not lactating. It is approximately 55 days until another attempt at insemination can be made (this number includes time to pregnancy check and time to return to oestrus). This means across the “influenced herd” there will be approximately 569,250 days (no RT), and 632,500 days without milk (with an RT), respectively. Taking into account the margin over purchased feed value of GBP 0.20 (see materials and methods), our calculations yielded a value of approximately GBP 5.40 per day, which we then multiplied by the days per year without milk. The other opportunity cost is the value of the calf that could be sold. The average value of a calf is around GBP 250; dividing that by the calving interval of 395 days gives a value of GBP 0.63 per day without a calf; this was also added to the days per year without milk. Combining these two values gives a value of around GBP 6.03 per day of opportunity cost to the farmer until the cow becomes pregnant. The final calculation in Table 2 leaves a cost of the days without milk at GBP 3,432,577.50 for herds inseminated by a bull without an RT and GBP 3,813,975 for herds inseminated by a bull with an RT—a difference of GBP 381,397.50 per year. This scenario pertains if the bull causes a 5% reduction in conception rate.

Unlike in pigs, a bull will stay in service for much longer, especially if he is of higher genetic merit and has a high worth [6]. That is, some bulls, if they are continuing to produce sufficient ejaculates, can stay in service up to 14 years, but 6 years is a reasonable average [6]. The longer a bull remains in service, the more costly it is to the producers. For example, if a bull with an RT were in service for 6 years, the cost would be GBP 2,291,745 and for 14 years, the costs would add up to around GBP 5,339,065 (Table 2). Conservative estimates are purposefully given in these calculations; clearly, if we impute a 10% reduction in the conception rate as a result of the RT then the final calculation of the differential in opportunity cost is much greater. If RECTs are present and the bull carrying them used for breeding, then the effect on conception rate is more likely to be nearer 10%, leading to an opportunity cost of GBP 1,201,117 per year (GBP 7,206,702 over 6 years, GBP 16,815,638 over 14 years).

**Table 2 animals-12-01982-t002:** Calculations leading to relative opportunity cost of a bull with an RT assuming 5% reduction in conception rate as reported for Rob1:29 [9,10] and 10% reduction in *italics* if all RTs are taken into account.

	Conception Rate	Successful Live Births per Year ^a^	Cows not Impregnated and Thus Not Lactating per Year ^a^	Days per Year without Milk ^b^	Opportunity Cost p.a. (Expressed as Value of Milk Lost GBP) ^c^	Opportunity Cost p.a. (Expressed as Value of Calf That Could Be Sold GBP) ^d^	Total Opportunity Cost per Year from Days without Milk (GBP) ^e^
Without RT	55%	12,650	10,350	569,250	3,073,950.00	358,627.50	3,432,577.50
With an RT	50%	11,500	11,500	632,500	3,415,500.00	389,475.00	3,813,975.00
	*45%*	*10,350*	*12,650*	*695,250*	*4,195,372.00*	*438,322.50*	*4,195,372.50*
Difference (i.e., RT cost)	5% ^f^	1150	1150	63,250	339,550.00	41,847.50	381,397.50
	*10% ^g^*	*2300*	*2300*	*126,000*	*1,121,422.00*	*79,695.00*	*1,201,117.00*

^a.^ Assuming 23,000 matings; ^b.^ Assuming 55 days until another attempt at insemination. ^c.^ Calculated as GBP 5.40 per day as follows: 4 months is subtracted from a total calving index of 13 months to account for dry and non-peak times for milking, meaning that we used a calving index of 9 months in this calculation as an estimate of the persistency of lactation, cows milked three times per day will on average lose 6% of their peak milk per month. Using a peak of 50 L, if we take this 9-month value into account, this gives a reduction in milk yield of 54% (27 L). Multiplying this value in time by the margin over purchased feed value works out at GBP 0.20 per litre milk produced. 27 L × 0.20 = 5.40 multiplied by days per year without milk ^b^. ^d^^.^ Calculated as GBP 0.63 per day based on the average value of a calf being GBP 250 and dividing that by the calving interval of 395 days, then multiplied by days per year without milk ^b^. ^e.^ Calculated as GBP 6.03 per day ^(c + d)^. ^f.^ Note, if only RobTs (or 1:29 alone) were screened then only a proportion of the benefits would be realized. ^g.^ Note, RECTs lead to a smaller proportion of chromosomally normal gametes than RobTs meaning that the impact on conception rate of RECTs is greater.

### 3.3. Relative Incidence of RobTs vs. RTs

The above calculations pertain to the effect of a Rob1:29 causing a 5% reduction in conception rate; however, another purpose of this study was also to establish the relative benefit of using our own screening approach (which will detect all RTs), compared to standard chromosome analysis (which will only detect RobTs) or a targeted approach that will detect Rob1:29 alone. In order to estimate the relative incidences of RECTs compared to RobTs, a number of sources were taken into account:The reported incidence of the Rob1:29 in the literature (between 0.4% and 60%) [9,10].The reported incidence of other RobTs (non 1:29) in the literature (between 0.03% and 6.4%), but bearing in mind that some of these would have been erroneously reported as 1:29, meaning that this incidence is probably higher [9,10].The reported incidence that RECTs are 4Õ more common than non-1:29 RobTs but bearing in mind that it is estimated that around 86% have remained undetected [11].The fact that no screening programme would have actively removed RECTs from a breeding herd in the way that they have been near eliminated for RobT 1:29 in Holstein cattle [12].

Clearly, the range of possibilities of the true incidence of RECTs by this calculation would range from less than 1% to over 100%—the former being unlikely and the latter being impossible. If we used a series of reasonable assumptions, however, namely that the overall incidence of all RobTs is 5% (if there is some active elimination of Rob1:29 from the herd), 10–25% of these are of other (non 1:29) chromosomes, the incidence of identifiable RECTs is four times that of non-1:29 RobT and 86% of RECTs remain undetected, a very conservative estimate of true incidence of RECTs suggests that they are *at least twice as prevalent as RobT 1:29 and probably much more so,* meaning that our test would be expected to detect at least 3x the total number of abnormalities than screening for Rob1:29 alone.

The system we developed [3] (and reported herein) detects all RTs (RobT and RECT); thus, the cost to a producer (and hence the benefits screening for RTs) depends on the frequency of the RTs in the pool of bulls used for insemination. If we assume that the frequency of RECTs is twice that of Rob1:29 (see above), then the benefits of screening for all RTs (as opposed to just RobTs or just Rob1:29 alone) are improved several-fold. A further consideration is that a RECT theoretically produces a greater proportion of chromosomally abnormal sperm than a RobT [13]; a RECT would therefore (again theoretically) reduce the conception rate more than Rob1:29, perhaps beyond 10%. Finally, the relative costs of the tests must be taken into account. Through personal contacts, we understand a karyotype (to detect RobTs only) typically costs around GBP 100 per bull (personal communication, various companies) compared to around GBP 300 for the approach described here. The example of financial impact calculated in this study (see Table 2 and described above) showed an RT bull with 23,000 matings per year, reducing conception rate by 5%, having a cost to producers of GBP 381,397.50 per year, or GBP 2,291,745.00 over a 6-year period. If we take the lowest reported incidence of Rob1:29 (0.4%–1 in 250 bulls), then, in this model, by screening for Rob1:29 only, it costs GBP 25,000 (GBP 100 × 250) on average to realise a benefit of GBP 2,291,745; a net gain of GBP 2,266,745 (around 2.8 million dollars). If we assume that RECTs are at least twice as common as Rob1:29, leaving a total RT incidence of 1.2% (1 in 83.33 bulls), and that the reduction in conception rate of a RECT is nearer 10%, then the cost to realise the benefits of identifying (on average) one bull is also GBP 25,000 (GBP 300 × 83.33), but for a much greater benefit (GBP 1,201,117 per year, GBP 7,206,702 over 6 years—a net gain of GBP 7,181,702 (nearly 9 million dollars). Higher incidences of RECTs and more severe reductions in conception rates increase this margin further. Moreover, these calculations do not take into account environmental damage and reputation loss caused by the use of hypoprolific bulls.

## 4. Discussion

So called “clinical” cytogenetics of domestic species have a long and distinguished history [12]. In pigs, activity was it its highest in the late the 20th century, with numerous cytogenetic screening programmes (notably the National Veterinary School of Toulouse in France) routinely screening samples. An unfortunate decline in the number and activity of screening laboratories, however, has not been accompanied by a decline in the problem. We recently suggested that around 1% of AI breeding boars currently carry an RT, twice that reported in the literature [3]. In sheep, goats, camels, chicken, duck and turkey, screening programmes largely do not exist, presumably due to the difficulty in interpreting the karyotype. While the cattle karyotype is similarly intractable (large number of similar looking chromosomes), the much higher value of breeding bulls (especially for dairy) compared to other species means that attempts to screen, at least for RobTs, remains paramount. In Sweden, the fertility of the whole cattle breeding population has demonstrably been increased through systematic screening and eradication of Rob1:29 and, in the UK and Australia, cytogenetic evaluation is a requirement for imported cattle [13]. RTs are heritable and transmitted to around half of the surviving offspring of the carrier. Therefore, the continuation of screening programmes is essential, even if the problem is near eradicated, as de novo abnormalities will inevitably occur, accumulate and ultimately be inherited.

Despite the clear association between hypo-prolificacy and all RTs, RECTs are not screened for in cattle, as a readily workable solution to screen for them has not hitherto been developed. In this paper, we provide a solution to redress this anomaly. As mentioned in the results section, RECTs, theoretically, carry a greater chance of reduction in productivity than RobTs, as the quadrivalent formed at meiosis impedes the process more, and leads to a greater proportion of chromosomally unbalanced products than the trivalent formed via a RobT [11]. For this reason, our calculations of financial loss were based on established figures of reduction in productivity of 5% established through the study of Rob1:29 (the most conservative end of the 5–10% figure quoted), and 10% for RECTs, although the effect is most probably greater still. Indeed, RECTs typically lead to a 50% reduction in litter size in pigs [7,8].

Even in animals such as humans and pigs where the karyotype is relatively easy to analyse, many translocations (regardless of whether they involve exchange of large blocks of chromatin) are difficult to identify by classical cytogenetics [7]. Moreover, some (cryptic) translocations cannot be detected by banding, regardless of preparation quality and tractability of karyotype [7]. The screening approach described herein, however, detects all RobTs and RECTs and analysis is both simple and straightforward to implement. Under the stated assumptions, our figures are consistently conservative and they do not take into account any potential loss in reputation or customer base that a breeder might face as a result, nor the costs involved with any attempt to replace the “missing” animals (if at all possible, due to the likely higher genetic merit of the new generation). It is also interesting to note that our calculations highlight how the more efficient a breeding business becomes (i.e., improved birth rates, less miscarriage, lower mortality) the more severe the economic impact of an RT will be. Across the global cattle breeding industry, breeding practices, average herd performance and monetary values inevitably display a large variability. As such, calculations such as the ones presented here can only give an idea of the order of magnitude of a particular problem and a tailored scenario calculation that will accurately reflect the economic impact of RTs for any specific breeder will require imputation of the producer’s specific needs into our algorithm. In our previous porcine study [8] we calculated likely losses of five scenarios in the pig breeding industry that, broadly speaking, encompassed a wide range of possible real-life cases that could be adapted to model any actual occurrence. Here, for cattle, we provide the framework through which the financial benefits of RT screening can be calculated, taking into account individual circumstances. We encourage companies to do this for their own particular purposes and all calculation spreadsheets are freely available by contacting the authors.

In pigs, the reported incidence of RTs is 0.47% [16]; however, our efforts with a multi-probe screening device suggests that the true incidence is probably around twice that number (around 1%) [8]. In bulls, the reported incidence of a single (Rob1:29) translocation is between 0.4% in Holstein cattle (in herds where there have been active efforts to eliminate it) to 60% (where they have not) [9,10]. In the results section, we make an attempt to perform the near impossible task of calculating the relative contribution of RECTs. A figure of twice the frequency of RECTs vs. RobTs (so our screening approach detects 3x as many abnormalities overall) does not seem unreasonable. In reality, it probably is much higher (and of course infinitely if all the RobTs have been eradicated). Therefore, the added ability to detect all RECTs, including the cryptic ones, suggests that a switch from the classical Giemsa staining method (that detects RobTs only) or PCR-based “Rob1:29 only” to widespread adoption of our multi-FISH approach should become a priority.

Finally, while this paper has concentrated on the financial impact of genetic abnormalities, the environmental consequences deserve consideration. A saving in wasted animals and resources also has a corresponding saving in methane or other greenhouse gas emissions. As we develop more sophisticated financial models to map the impact of RTs, future studies will concentrate on their environmental impact also likely incorporating methodologies such as lifecycle analysis.

## 5. Conclusions

Breeding companies in dairy or meat are tasked with supporting the increasing global nutritional demand, increasing profitability, reducing waste and environmental damage while simultaneously ensuring high levels of animal welfare. In 2020, the global cattle production industry exceeded a total net worth of USD 827.4 billion for dairy and USD 467.7 billion for beef. Our work in pigs has shown that RTs persist in herds, despite widespread eradication efforts [16], and we see no reason why this should not be the case for cattle for all RTs, not just Rob1:29. RobTs are particularly prone to de novo occurrence in acrocentric chromosomes due to the similarity in centromeric sequences and (the X chromosome aside) the cattle genome consists of nothing but acrocentric chromosomes. Without an efficient and proactive screening programme, such as using the approach outlined in this paper, the negative effects of RTs (both RobT and RECT) will continue to impact the global cattle industry; currently, the differences among bulls are usually only appreciated after the bull has produced many offspring, by which time it may be reaching the end of its reproductive life. Finally, the results herein represent a hypothetical scenario based on the assumptions of a Rob1:29 incidence of 0.4%, impacting conception rates by 5%, and all RTs with an incidence of 1.2%, impacting conception rates by 10%. We believe that these are conservative estimates, and although the numbers are calculated to the nearest GBP (such is the nature of financial models), the precise numbers are not important. What is more important is that we have developed a means to calculate financial benefits (into which specific figures can be imputed) and, more important still, developed a solution for eradicating all RTs.

## Figures and Tables

**Figure 1 animals-12-01982-f001:**
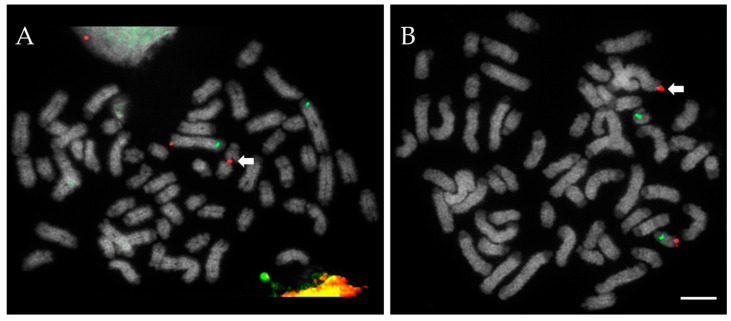
(**A**) Probes for proximal 1q (green) and distal 1q (red) both correctly hybridize to the normal chromosome 1; however, a translocation is detected due to the distal q-arm signal localizing on chromosome 25 instead (arrow). (**B**) Probes for proximal 25q (green) and distal 25q (red) both correctly hybridize to the normal chromosome 25; however, a translocation is detected due to the distal q-arm signal localizing on chromosome 1 instead (arrow). Together, the two images (**A**,**B**) establish this was a case of reciprocal translocation (RECT) t (1:25). DNA counterstained in DAPI (grey). Scale bar = 5 µm.

**Table 1 animals-12-01982-t001:** Chromosomal errors detected by FISH screening among *n* = 59 individuals. RobT = Robertsonian translocation; RECT = Reciprocal translocation. * Chimeric bull 10 cell analysed 50:50 XX/XY.

Karyotype	Total Number of Cases
Heterozygous RobT (1:29)	5
Homozygous RobT (1:29)	2
RECT (1:25)	1
RECT (12:23)	2
Complex Translocation (26)	1
XX/XY chimeric *	1
Normal	47

## Data Availability

All spreadsheets available by contacting the authors.

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
