# Peer review of "The Economic Burden of Chromosome Translocations and the Benefits of Enhanced Screening for Cattle Breeding"

_animals, 2022, doi:10.3390/ani12151982_

Round 1

Reviewer 1 Report

This is an interesting, important, and useful paper.

My main concern is the insufficient description of the material of the cytogenetic study.

What was the age of the bulls?

To which breed belonged the carriers of the translocations?

Was there a relationship between those who carried the same aberration?

How many cells per sample were analyzed?

What was the number of false negative and false positive cells?

 What was the percentage of the cell containing XX and XY in the chimeric bull?

I did not find the Institutional Review Board Statement and approval number for studies involving animals, which is necessary  according to the Instructions for Authors. The statement "We are grateful to the companies who provided cattle blood samples for these studies" does not seem sufficient.

One technical comment about the Fig. 1 caption.

In the electronic media, the statement "Total magnification 1000×" is meaningless. It should be replaced by the scale bar.

Author Response

This is an interesting, important, and useful paper. THANKS YOU FOR THE KIND WORDS

My main concern is the insufficient description of the material of the cytogenetic study. WE HOPE WE HAVE RECTIFIED THIS

What was the age of the bulls? WE DO NOT GET SPECIFIC INFORMATION FOR EACH BULL SENT, HOWEVER OUR INFORMATION IS THAT THEY ARE ALL 2 TO 6 YEARS - A NOTE TO THIS EFFECT HAS BEEN ADDED (line 149-151)

To which breed belonged the carriers of the translocations? WE HAVE ADDED THIS INFORMATION (line 149-151)

Was there a relationship between those who carried the same aberration? NOT TO OUR KNOWLEDGE (line 241)

How many cells per sample were analyzed? 3-5 METAPHASES (line 174-175)

What was the number of false negative and false positive cells? I AM NOT SURE HOW WE WOULD DETERMINE THIS.  ASIDE FROM THE XX/YY ALL CELLS ANALYSED SHOWED THE SAME KARYOTYPE

 What was the percentage of the cell containing XX and XY in the chimeric bull? 50:30 - NOW IN FIGURE LEGEND (line 247)

I did not find the Institutional Review Board Statement and approval number for studies involving animals, which is necessary  according to the Instructions for Authors. The statement "We are grateful to the companies who provided cattle blood samples for these studies" does not seem sufficient. AS ALL SAMPLES WERE TAKEN FOR ROUTINE TESTING, UNIVERSITY OF KENT RESEARCH ETHICS COMMITTEE HAVE ADVISED THAT NO FURTHER ETHICAL OVERSIGHT IS REQUIRED. A NOTE TO THIS EFFECT HAS BEEN ADDED (line 149-150)

One technical comment about the Fig. 1 caption.

In the electronic media, the statement "Total magnification 1000×" is meaningless. It should be replaced by the scale bar. WE HAVE CHANGED THIS TO A 5um SCALE BAR - THANK YOU FOR THE COMMENT (line 252-254)

Reviewer 2 Report

This is an extremely well written and carefully prepared manuscript that describes a novel and innovative study that has both basic science and and applied applications in the agriculture sector. The methodology for the the detection and assessment of chromosome abnormalities in cattle is novel and based on a previously developed swine protocol in the team's laboratory.  Aspects of the the procedure are proprietary. Ir would have been interesting to know how many metaphases per "square"  were examined, and in the case of the XX/XY chimera if the proportion on XX cells corresponded to that detected with conventional chromosome staining. While not a necessary addition the authors may wish to consider commenting on this.

While I am not an economist, the approach and results relate to the potential economic impact to the dairy and beef industry are logical and conclusions and wile they reflect the impact in current economic time   they are in keeping with the previous recognition that undetected chromosome abnormalities can severely impact productivity and cause financial loss.

Author Response

Many thanks for these kind words. The 2 issues are addressed. Lines 174-5 and 247